# Non-Invasive Assessment of Isocitrate Dehydrogenase-Mutant Gliomas Using Optimized Proton Magnetic Resonance Spectroscopy on a Routine Clinical 3-Tesla MRI

**DOI:** 10.3390/cancers15184453

**Published:** 2023-09-07

**Authors:** Laiz Laura de Godoy, Kheng Choon Lim, Archith Rajan, Gaurav Verma, Mauro Hanaoka, Donald M. O’Rourke, John Y. K. Lee, Arati Desai, Sanjeev Chawla, Suyash Mohan

**Affiliations:** 1Department of Radiology, Perelman School of Medicine at the University of Pennsylvania, Philadelphia, PA 19104, USA; laiz.godoy@pennmedicine.upenn.edu (L.L.d.G.); archith.rajan@pennmedicine.upenn.edu (A.R.); mauro.hanaoka@pennmedicine.upenn.edu (M.H.); suyash.mohan@pennmedicine.upenn.edu (S.M.); 2Department of Neuroradiology, Singapore General Hospital, Singapore 169609, Singapore; lkchoon@gmail.com; 3Department of Radiology, Icahn School of Medicine at Mount Sinai, New York, NY 10029, USA; mahagaurav@gmail.com; 4Department of Neurosurgery, Perelman School of Medicine at the University of Pennsylvania, Philadelphia, PA 19104, USA; donald.orourke@pennmedicine.upenn.edu (D.M.O.); john.lee3@pennmedicine.upenn.edu (J.Y.K.L.); 5Abramson Cancer Center, Perelman School of Medicine at the University of Pennsylvania, Philadelphia, PA 19104, USA; arati.desai@pennmedicine.upenn.edu; 6Glioblastoma Translational Center of Excellence, Perelman School of Medicine at the University of Pennsylvania, Philadelphia, PA 19014, USA

**Keywords:** 2-hydroxyglutarate, isocitrate dehydrogenase, proton MR spectroscopy, glioma

## Abstract

**Simple Summary:**

*IDH* mutation is one of the most important prognostic biomarkers in glioma management. Noninvasive neuroimaging techniques to predict *IDH* mutant glioma may be valuable for guiding clinical decision-making and monitoring response to targeted therapies. The aim of our prospective study was to investigate the clinical potential of proton MR spectroscopy (^1^H-MRS) with an optimized TE (97 ms) in identifying *IDH*-mutant gliomas by detecting characteristic resonances of 2HG and its complex interplay with other clinically relevant metabolites. We confirmed that the oncometabolite 2HG was found to be significant in predicting *IDH*-mutant gliomas, and both single-voxel and multi-voxel ^1^H-MRS methods are equally efficient in detecting complex resonances of 2HG. Additionally, Glx (glutamate + glutamine) and NAA (N-acetylaspartate) were also found to be important in distinguishing *IDH*-mutant from wild-type gliomas. In short, ^1^H-MRS with an optimized TE may be helpful for noninvasively detecting the abnormally high levels of 2HG with high accuracy and comprehending its interaction with other relevant metabolites in infiltrative gliomas.

**Abstract:**

Purpose: The isocitrate dehydrogenase (*IDH*) mutation has become one of the most important prognostic biomarkers in glioma management, indicating better treatment response and prognosis. *IDH* mutations confer neomorphic activity leading to the conversion of alpha-ketoglutarate (α-KG) to 2-hydroxyglutarate (2HG). The purpose of this study was to investigate the clinical potential of proton MR spectroscopy (^1^H-MRS) in identifying *IDH*-mutant gliomas by detecting characteristic resonances of 2HG and its complex interplay with other clinically relevant metabolites. Materials and Methods: Thirty-two patients with suspected infiltrative glioma underwent a single-voxel (SVS, n = 17) and/or single-slice-multivoxel (^1^H-MRSI, n = 15) proton MR spectroscopy (^1^H-MRS) sequence with an optimized echo-time (97 ms) on 3T-MRI. Spectroscopy data were analyzed using the linear combination (LC) model. Cramér–Rao lower bound (CRLB) values of <40% were considered acceptable for detecting 2HG and <20% for other metabolites. Immunohistochemical analyses for determining *IDH* mutational status were subsequently performed from resected tumor specimens and findings were compared with the results from spectral data. Mann–Whitney and chi-squared tests were performed to ascertain differences in metabolite levels between *IDH*-mutant and *IDH*-wild-type gliomas. Receiver operating characteristic (ROC) curve analyses were also performed. Results: Data from eight cases were excluded due to poor spectral quality or non-tumor-related etiology, and final data analyses were performed from 24 cases. Of these cases, 9/12 (75%) were correctly identified as *IDH*-mutant or *IDH*-wildtype gliomas through SVS and 10/12 (83%) through ^1^H-MRSI with an overall concordance rate of 79% (19/24). The sensitivity, specificity, positive predictive value, and negative predictive value were 80%, 77%, 86%, and 70%, respectively. The metabolite 2HG was found to be significant in predicting *IDH*-mutant gliomas through the chi-squared test (*p* < 0.01). The *IDH*-mutant gliomas also had a significantly higher NAA/Cr ratio (1.20 ± 0.09 vs. 0.75 ± 0.12 *p* = 0.016) and lower Glx/Cr ratio (0.86 ± 0.078 vs. 1.88 ± 0.66; *p* = 0.029) than those with *IDH* wild-type gliomas. The areas under the ROC curves for NAA/Cr and Glx/Cr were 0.808 and 0.786, respectively. Conclusions: Noninvasive optimized ^1^H-MRS may be useful in predicting *IDH *mutational status and 2HG may serve as a valuable diagnostic and prognostic biomarker in patients with gliomas

## 1. Introduction

Since the fourth edition of the WHO Classification of Tumors of the Central Nervous System in 2016, isocitrate dehydrogenase (*IDH*) mutations have become one of the most important prognostic biomarkers in adult-type infiltrative gliomas [1]. *IDH* mutational status can distinctly separate astrocytomas and oligodendrogliomas from more aggressive and deadly glioblastomas, regardless of histopathological features [2]. It is well recognized that astrocytomas and oligodendrogliomas harboring the *IDH* mutation demonstrate a better response to chemoradiation therapy, and these patients generally demonstrate better survival outcomes than those with glioblastomas harboring *IDH* wild-type alleles [3,4], thus emphasizing the importance of non-invasive identification of *IDH* mutant gliomas.

*IDH* mutations gain a neomorphic enzyme activity leading to the conversion of alpha-ketoglutarate (α-KG) to 2-hydroxyglutarate (2HG) in the citric acid cycle, resulting in abnormally high levels of 2HG (Figure 1) [5]. The oncometabolite 2HG has been proposed as a surrogate biomarker of *IDH* mutational status in gliomas. Proton magnetic resonance spectroscopy (^1^H-MRS) allows non-invasive assessment of the metabolic landscape of biological tissues [6,7,8,9]. Several prior studies have reported the potential utilities of ^1^H-MRS in diagnosis, planning treatment strategies, and assessing treatment response in gliomas [10,11,12]. However, precise in vivo detection of 2HG on clinical magnetic field strengths (3T) demands the development of optimized ^1^H-MRS sequences due to the extensive overlap of 2HG resonances with that of neighboring metabolites such as glutamate, glutamine, and γ-aminobutyric acid (GABA) [12]. A spin-echo point-resolved spectroscopy (PRESS) sequence with an optimized echo time (TE) of 97 ms has been proposed to identify a well-defined narrow 2HG peak at 2.25 ppm, resulting in the detection and quantification of 2HG with greater sensitivity [13]. 

While immunohistochemical analyses and exomic sequencing are considered gold standards for determining *IDH* mutational status in gliomas [14,15], tissue heterogeneity, partial sampling of tissue specimens, and the presence of variable amounts of antigens constrain the utility of these methods in the reliable detection of *IDH* mutational status [16]. Eloquent locations of these neoplasms may also limit neurosurgical interventions. Therefore, the non-invasive identification of *IDH* mutant gliomas using ^1^H-MRS plays an important role in patient counseling for therapeutic intervention and prognostication and could be utilized as a pharmacodynamic indicator to monitor treatment response [17,18]. 

We hypothesized that the pathophysiologic changes due to *IDH* mutations leading to the accumulation of oncometabolite 2HG could also cause variations in the neurochemical profile of other metabolites observed on ^1^H-MRS. Therefore, the purpose of this study was to investigate the clinical potential of ^1^H-MRS, single-voxel (SVS), and single-slice-multivoxel magnetic resonance spectroscopic imaging (^1^H-MRSI) in identifying *IDH*-mutant gliomas by detecting characteristic resonances of 2HG and its complex interplay with other clinically relevant metabolites.

## 2. Methods

### 2.1. Subjects

A waiver for the informed consent form was requested at the time of Institutional Review Board (IRB) submission, and the research project was approved by the *IRB/*Ethics Committee of the University of Pennsylvania (protocol # 829645). This study was designed and conducted according to the guidelines of the Declaration of Helsinki. A total of 32 patients, comprising of 30 suspected newly diagnosed infiltrative gliomas and two suspected neoplastic progression, were recruited based on routine MRI findings for this study.

### 2.2. Data Acquisition

All patients underwent anatomical imaging, single-voxel (SVS, n = 17), and/or single-slice-multivoxel magnetic resonance spectroscopic imaging (^1^H-MRSI, n = 15) MRI on a 3T Tim Trio whole-body MR scanner (Siemens, Erlangen, Germany) equipped with a 12-channel phased-array head coil. The anatomical imaging protocol included axial 3D-T1-weighted magnetization-prepared rapid acquisition of gradient echo (MPRAGE) imaging and an axial T2-weighted fluid-attenuated inversion recovery (FLAIR) imaging using standard parameters. The postcontrast T1-weighted images were acquired with the same parameters as the precontrast acquisition after administration of a standard dose (0.14 mmol/Kg) of gadobenate dimeglumine (MultiHance, Bracco Imaging, Milano, Italy) intravenous contrast agent using a power injector (Medrad, Idianola, PA, USA).

Both SVS and ^1^H-MRSI were acquired using a standard spin-echo point-resolved spectroscopy (PRESS) sequence. Sequences parameters for SVS included repetition time (TR)/echo time (TE)/number of excitations (NEX) = 2000/97 ms/128, bandwidth = 1200 Hz. Depending upon the tumor location, size, and geometry and to minimize the prospect of obtaining spectral data with poor signal-to-noise ratio and partial volume-averaging effects, the single voxel size varied from 10 × 10 × 10 mm^3^ (volume: 1 cm^3^) to 20 × 20 × 20 mm^3^ (volume: 8 cm^3^). Sequences parameters for ^1^H-MRSI included TR/TE/NEX = 2000/97 ms/3, bandwidth = 1200 Hz, field of view = 16 × 16 cm^2^–20 × 20 cm^2^, slice thickness = 15–20 mm, and matrix size = 16 × 16. The typical voxel size varied from 10 × 10 × 15 mm^3^ (volume: 1.5 cm^3^) to 10 × 12.5 × 20 mm^3^ (volume: 2.5 cm^3^) depending upon the dimensions of the neoplasms. For ^1^H-MRSI, the volume of interest (VOI) was selected to include neoplasms visible as a hyperintense mass on T2-FLAIR images and contralateral normal brain parenchyma, avoiding scalp, skull base, or sinuses. The outer-volume saturation slabs were placed outside VOI to suppress lipid signals from the scalp. The saturation slabs were applied along multiple directions, depending on tumor location, morphology, and orientation (Figure 2). The data set was acquired using elliptical-K-space sampling with weighted phase encoding. Manual shimming was performed to achieve an optimal fullwidth at half-maximum value (<20 Hz) of the magnitude water signal. Both water-suppressed and unsuppressed spectra were acquired, and the unsuppressed water signal was used for computing metabolite concentrations.

### 2.3. Data Processing

All spectroscopy data were analyzed using a user-independent spectral fit program (linear combination (LC) model, (http://s-provencher.com/lcmodel.shtml) [19]. The region between 0.2 and 4.2 ppm of the spectrum was analyzed, and the following metabolites were evaluated: Lipids + Lactate, 1.3 ppm; N-acetylaspartate (NAA), 2.02 ppm; creatine (Cr), 3.02 ppm; choline (Cho), 3.22 ppm; glutamate+glutamine (Glx), 2.24 to 2.34 ppm; myoinositol (mI), 3.56 ppm; and 2HG, 2.25 ppm. The resonance of Cr at 3.02 ppm was used as an internal chemical shift reference for computing metabolite ratios. The quality of spectral fitting was evaluated by analyzing the difference spectrum (fitted spectrum subtracted from the original spectrum) and by using Cramér–Rao lower bound (CRLB) values, with less than 40% considered acceptable for detecting 2HG [20] and less than 20% for all other metabolites [21,22]. The number of voxels analyzed in the cases of ^1^H-MRSI varied from 2 to 41, and at least two neighboring voxels with CRLB values <40% for 2HG were required to classify the cases as 2HG-positive. Histopathological and immunohistochemical analyses for glioma-grade and *IDH* mutational status were subsequently performed from resected tumor specimens (complete or near total resection of the hyperintense expansile mass as visible on T2-FLAIR image), and the findings were compared with the results from spectral data. Metabolic ratios for NAA/Cr, Cho/Cr, Glx/Cr, mI/Cr, and Lipids + Lactate/Cr from the selected voxels were computed from each patient. Additionally, the absolute concentration of 2HG and metabolite ratio of 2HG/Cr were computed from true-positive *IDH* mutant gliomas.

### 2.4. Determination of IDH Status Using Immunohistochemistry

Tumor specimens were fixed in formalin and processed for paraffin embedding. Hematoxylin and eosin staining and immunohistochemistry were conducted on 5-micron-thick formalin-fixed, paraffin-embedded tissue sections mounted on Leica Surgipath slides followed by drying for 60 min at 70 °C. Immunohistochemistry with the anti-*IDH*1-R132H antibody (Monoclonal Mouse Anti-human *IDH*1 (R132H), Dianova, DIA Clone H09) and DAB chromogen was performed on a Leica Bond III instrument using Bond Polymer Refine Detection System (Leica Microsystems AR9800) following a 20 min heat-induced epitope retrieval with Epitope Retrieval 2, EDTA, pH 9.0 [23].

### 2.5. Statistical Analysis

Statistical analyses were performed using a statistical package, SPSS for Windows (v. 18.0; Chicago, IL, USA). Kolmogorov–Smirnov tests were used to determine the nature of data distribution. As the data showed a departure from Gaussian distribution, non-parametric Mann–Whitney U tests were performed to assess differences in metabolite levels between *IDH*-mutant and *IDH*-wild-type gliomas. The chi-square test was used to assess differences in categorical variables. A probability (*p*) value of less than 0.05 was considered significant. Sensitivity, specificity, positive predictive value, and negative predictive value were determined. Receiver operating characteristic (ROC) curve analyses were also performed for the metabolites exhibiting significant differences between two groups (*IDH*-mutant and *IDH*-wild-type gliomas).

## 3. Results

Of these 32 patients, eight were excluded as five patients had sub-optimal spectral quality due to inadequate water suppression for the reliable detection and quantification of metabolites, and three other patients were excluded because they had final histological diagnoses of non-tumor etiology, with pathology reports consistent with reactive brain changes without molecular features of neoplasms. As such, a total of 24 patients were included (mean age = 48.7 ± 15 years, 11 males and 13 females) in the final data analysis (Figure 3—Flowchart). All cases had histopathological confirmation for infiltrative gliomas (22 newly diagnosed and 2 neoplastic progression), ranging from histological grade 2 to grade 4. The immunohistochemistry analyses revealed 15 *IDH*-mutant gliomas and 9 *IDH*-wild type glioblastomas. Patient demographics, along with histopathological grading and immuno-histochemical findings, are presented in Table 1.

While SVS was preferred for acquiring spectroscopy data from focal well-circumscribed, more ‘superficial’ lesions, ^1^H-MRSI was chosen for ill-defined, irregular heterogeneous and larger lesions to avoid incomplete sampling of neoplasms. Because of time constraints, we were able to employ both techniques only in a few cases (n = 3). 

Using a CRLB <40% for the detection of 2HG, 9/12 (75%) cases were correctly identified as *IDH-*mutant or *IDH-*wild-type gliomas by SVS and 10/12 (83%) by ^1^H-MRSI (Figure 4 and Figure 5, respectively), with an overall concordance rate of 79% (19/24). Of 5 patients who incorrectly classified by ^1^H-MRS for *IDH* mutational status, 2 were false positives (1 each on SVS and ^1^H-MRSI) and 3 were false negatives (2 on SVS, 1 on ^1^H-MRSI). The sensitivity, specificity, positive predictive value (PPV), and negative predictive value (NPV) in identifying *IDH*-mutant and *IDH*-wild-type gliomas were 80%, 77%, 86%, and 70%, respectively (Table 2). The mean concentration and standard errors of 2HG and metabolite ratio of 2HG/Cr in *IDH* mutant cases were 5.24 ± 1.59 mM and 0.55 ± 0.08, respectively. The metabolite 2HG was found to be significant in predicting *IDH*-mutant gliomas using chi-squared test (*p* < 0.01).

The *IDH*-mutant gliomas also harbored a significantly higher mean concentration of NAA/Cr ratio (1.20 ± 0.09 vs. 0.75 ± 0.12; *p* = 0.016) and significantly lower Glx/Cr ratio (0.86 ± 0.08 vs. 1.88 ± 0.66; *p* = 0.029) than those with *IDH* wild-type gliomas. There were no significant differences (*p* > 0.05) in the mean concentration for the remaining metabolite ratios between *IDH*-mutant and *IDH*-wild-type gliomas (Figure 6). The ROC analyses revealed that areas under the ROC curves for NAA/Cr and Glx/Cr were 0.808 and 0.786, respectively, in distinguishing *IDH-*mutant from *IDH* wild-type gliomas. 

## 4. Discussion

In this study, we prospectively analyzed the clinical utility of SVS and ^1^H-MRSI using an optimized TE (97 ms) in assessing *IDH*-mutational status by detecting the characteristic resonances of 2HG in patients presenting with newly diagnosed infiltrative gliomas and suspected neoplastic progression. Our results showed that ^1^H-MRS can identify *IDH*-mutant gliomas with high accuracy (79%), sensitivity (80%), and specificity (77%). We also found that the *IDH* mutation affects other metabolic pathways leading to variations in metabolite pools of Glx and NAA. These observations provide insights into the pathophysiology of *IDH* mutations in gliomas. Moreover, the accurate determination of *IDH* mutational status at the time of initial presentation has important therapeutic implications when a critical decision about the selection of the optimal treatment strategy is to be made.

Mechanistically, wild-type *IDH* normally catalyzes the reversible NADP+dependent oxidative decarboxylation of isocitrate to α-KG in the citric acid cycle, while *IDH* mutations confer a neomorphic enzyme activity converting α-KG to 2HG. The high levels of 2HG change the cellular metabolism by leading to DNA hypermethylation and epigenetic modifications of histone, resulting in tumorigenesis [5,24]. The oncometabolite 2HG has been proposed as a putative biomarker for *IDH*-specific genetic profiles for gliomas. However, not all *IDH*-mutant gliomas, especially the non-canonical *IDH* mutant gliomas (about 20–25% of grade-2 and 5–12% of grade-3 gliomas), show the neomorphic activity of 2HG production [25], suggesting that 2HG detection alone may not always be sufficient for identifying the *IDH* mutation in gliomas. By exploring the reasons for obtaining false-positive cases, we noted that small tumor volume was found to be a potential limiting factor for the reliable detection of 2HG resonances in our study. Similar findings were observed in a previous study in which investigators reported higher detection sensitivity of 2HG from larger voxel size [26]. 

The optimized TE allowed us to investigate specific biochemical alterations influenced by the pathologic production of 2HG in *IDH*-mutant gliomas, showing a clear advantage compared to routine short TE (30–35 ms) [18]. We could detect the 2HG peak at 2.25 ppm with high sensitivity and specificity (80% and 77%, respectively). Additionally, in line with our hypotheses and prior studies [27,28,29], the Glx (glutamate + glutamine)/Cr was found to be significantly decreased in *IDH*-mutant gliomas in the present study. Our results are supported by an earlier study reporting that glutamate levels become depleted by the enzymatic activity of glutamate dehydrogenase in an attempt to replenish α-KG lost in the conversion to 2HG by the *IDH* enzyme [30]. Metabolomic analyses using glioma cell lines and surgical specimens have also shown that glutaminolysis serves as a key compensatory pathway to maintain metabolic homeostasis in *IDH* mutant gliomas. As a result, glutamate levels are significantly reduced in *IDH* mutant gliomas compared to *IDH* wild-type counterparts [31]. Additionally, we observed significantly increased levels of NAA in the *IDH*-mutant group compared to the *IDH*-wild-type group. However, the biological mechanism of NAA and whether it contributes to tumor pathogenesis remains unclear, and this metabolite might be a confounder of tumor grade between low- and high-grade gliomas [32,33,34]. 

There is no established optimal CRLB cutoff value or concentration level for the detection of 2HG [18]. Using individual patient data and CRLB values, the optimal cutoff for 2HG was found to be lower than 40% in our cohort [20]. Further validation of this optimal threshold for 2HG is crucial for the application of glioma management in clinical practice. A benefit of 2HG MRS compared to other MRI techniques is that it can be employed as direct evidence of increased measurements of 2HG with enhanced detection accuracy [35]. On the other hand, conventional MRI techniques have shown a wide variability of sensitivities (71–100%) and specificities (51–100%) in the identification of *IDH* mutant gliomas [36,37,38]. Similarly, studies using diffusion-weighted imaging and perfusion-weighted imaging have also reported a wide range of sensitivities (56–100%) and specificities (63–100%) for distinguishing *IDH*-mutant from *IDH*-wild-type gliomas [18,39,40,41].

SVS is a broadly available method, generally providing a better signal-to-noise ratio especially from focal well-circumscribed lesions [42]. Moreover, SVS is an easy-to-acquire sequence that allows good magnetic field shimming over the limited volume of interest and takes shorter acquisition time. On the other hand, ^1^H-MRSI is a valuable technique for obtaining metabolic information from ill-defined, irregular, heterogeneous, large(er) masses including the entire tumor volume to avoid incomplete sampling. Additionally, the ^1^H-MRSI method allows acquisition of spectroscopy data with better spatial resolution, thus minimizing the partial volume-averaging effects [35]. This in turn allows the metabolic mapping of tumor heterogeneity with a lesser degree of admixing the signals from different tissue compartments. We believe that both methods are equally good when used appropriately according to the specific tumor’s features. However, ^1^H-MRSI takes a longer time to acquire the data and this sequence is prone to the voxel bleeding effect, resulting in signal contamination from lipids originating from the scalp and bone marrow. Moreover, it is more difficult to shim well over the larger VOI. 

Some previous studies have employed sophisticated spectroscopic sequences such as multiple quantum-filtered and spectral editing techniques and post-processing tools for unambiguously detecting spectral resonances of 2HG from *IDH* mutant gliomas [23,27,43,44]. Earlier, we have also shown the potential of using two-dimensional localized correlation spectroscopy (2D-L-COSY) at 7 Tesla to detect 2HG in *IDH*-positive gliomas [23]. However, these sophisticated spectroscopic sequences and tools are not readily and widely available in routine clinical settings.

^1^H-MRS evaluation of the oncometabolite 2HG is of clinical interest in creating a noninvasive detection technique for *IDH* mutant gliomas for diagnostic purposes; in differentiating glioma from other etiologies (i.e., solitary metastasis, demyelinating lesion) [22]; in surgical decision making, as aggressive resection of both enhancing and nonenhancing disease might improve survival in *IDH*-mutant, but not in *IDH* wild-type gliomas [45]; in treatment monitoring (concentration of 2HG increases sharply with tumor progression, whereas it decreases in response to radio- and chemotherapies) [46]; and even in differentiating tumor recurrence from treatment-related changes. In addition, a recent phase I clinical trial used 2HG MRS to document treatment response to a mutant-*IDH*1 inhibitor drug and revealed a significant decreasing rate of 2HG levels after one week of treatment [47]. Notably, treatment monitoring of novel target therapies using 2HG detection by MRS may be exploited for personalized and precision medicine and early treatment response in clinical trials.

Our findings may aid in further establishing 2HG as a surrogate marker of *IDH* mutational status in gliomas. Using an optimized TE of 97 ms, both SVS and multi-voxel ^1^H-MRS methods are equally efficient in detecting complex resonances of 2HG. Other physiologically sensitive metabolites such as Glx and NAA may also serve as potential biomarkers for detecting *IDH* mutant gliomas. We believe that the diagnostic performance of ^1^H-MRS in the identification of *IDH* mutational status in gliomas can be further improved by using an integrated approach of analyzing metabolite levels of 2HG, NAA, and Glx together in future studies. 

## 5. Limitations

Despite promising results, our study had some shortcomings, including a small patient size and the non-availability of *IDH*2 mutational status. Although we used a relatively high CRLB value of 40% for detecting 2HG, further in vivo and in vitro experiments are required to determine the optimal threshold CRLB value for detecting 2HG metabolite. Our findings warrant further validation in large-scale and multicenter prospective studies for developing a fast, reliable, and reproducible method for identifying *IDH* mutational status in gliomas.

## 6. Conclusions

^1^H-MRS with an optimized TE may be useful for noninvasively detecting the abnormally high levels of 2HG with high accuracy and understanding its interaction with other important metabolites in infiltrative gliomas. Our findings indicate that 2HG and Glx are potential noninvasive surrogate biomarkers for detecting *IDH* mutations, which has significant clinical implications for prognostication and implementation of appropriate clinical management procedures in glioma patients.

## Figures and Tables

**Figure 1 cancers-15-04453-f001:**
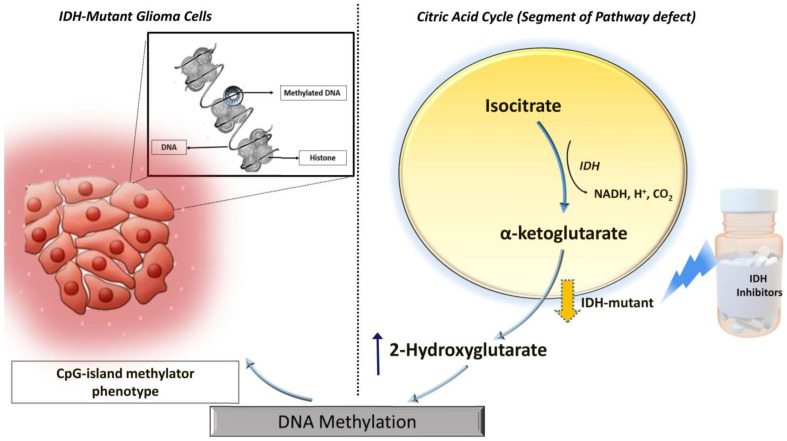
Relationship of *IDH* mutations to levels of 2HG (an oncometabolite) and DNA methylation.

**Figure 2 cancers-15-04453-f002:**
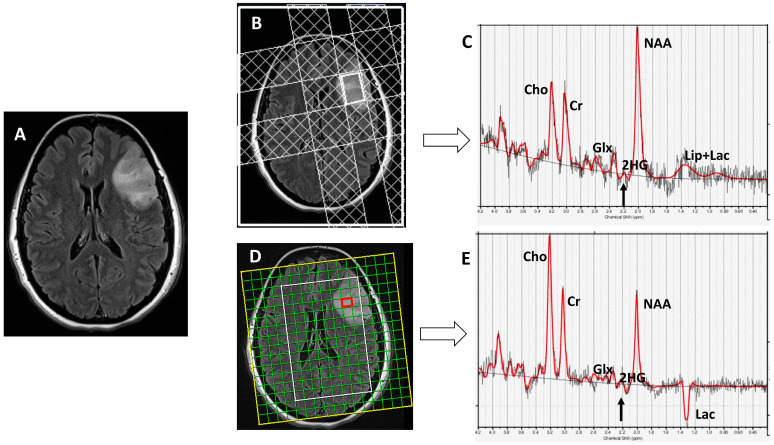
A patient with anaplastic astrocytoma. (**A**) T2-FLAIR shows a left frontal lobe mass. (**B**) Four outer-volume saturation slabs are placed outside the single voxel to suppress contamination from lipid signals arising from the scalp and bone marrow. (**C**) Spectra demonstrate elevated Cho/Cr (0.63; CRLB = 2%) and metabolic levels of 2HG (2HG/Cr = 0.4; CRLB = 33%; black arrow). (**D**) ^1^H-MRSI grid overlaid on T2-FLAIR image showing different voxels from tumor. (**E**) Spectra from a red voxel within the mass demonstrates elevated Cho/Cr (0.84; CRLB = 3%) and metabolic levels of 2HG (2HG/Cr = 0.49; CRLB = 36%, black arrow). Histopathological and immunohistochemical analyses were consistent with grade-3 astrocytoma with positive *IDH*1 mutational status. Red lines: fitting spectra. Black lines: real spectra.

**Figure 3 cancers-15-04453-f003:**
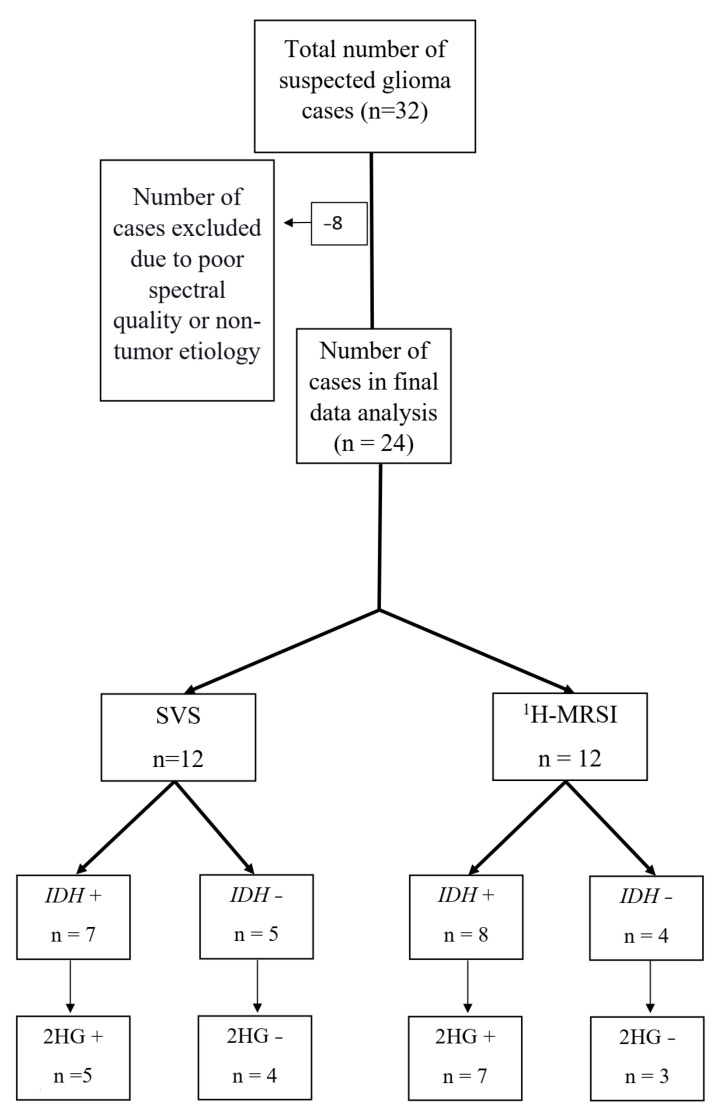
The flow-chart documenting inclusion and exclusion criteria along with number of patients who underwent SVS or ^1^H-MRSI and *IDH* mutational status of gliomas.

**Figure 4 cancers-15-04453-f004:**
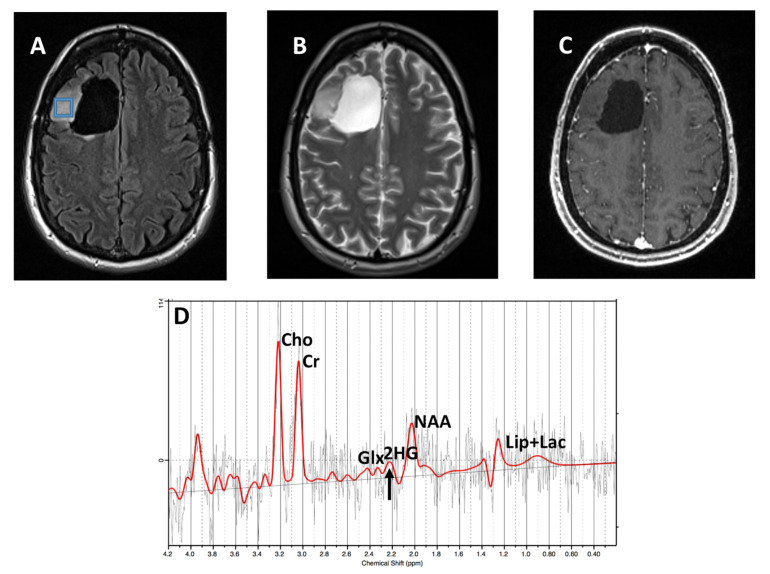
A glioma patient, status post resection. (**A**) T2-FLAIR and (**B**) T2-W images demonstrate expansile signal abnormality lateral to resection cavity in the right frontal lobe. (**C**) Post-contrast T1 image shows no abnormal enhancement. (**D**) SVS demonstrates elevated Cho/Cr (0.37; CRLB = 6%) and metabolic levels of 2HG (2HG/Cr = 0.4; CRLB = 33%; black arrow). Histopathological and immunohistochemical analyses were consistent with grade-3 oligodendroglioma with positive *IDH*1 mutational status. Red lines: fitting spectra. Black lines: real spectra.

**Figure 5 cancers-15-04453-f005:**
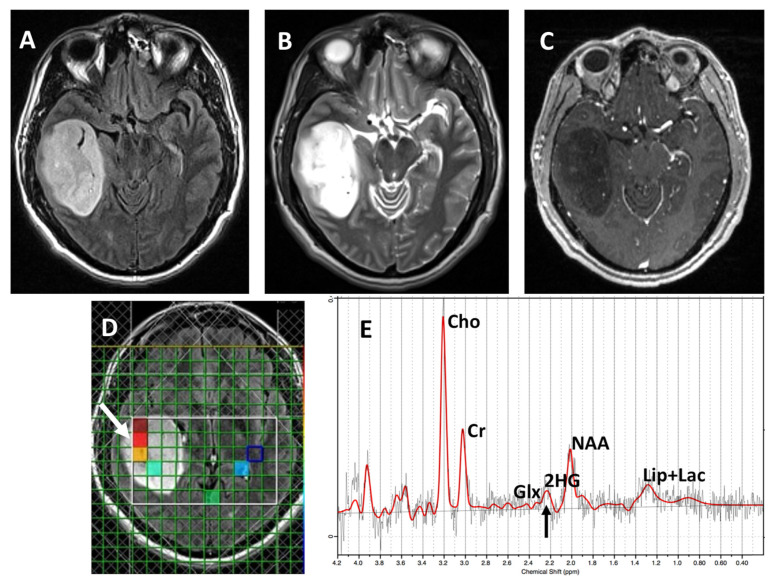
A patient with anaplastic astrocytoma. (**A**) T2-FLAIR and (**B**) T2-weighted image show a right temporal lobe mass. (**C**) Post-contrast T1 image shows faint enhancing foci within the mass. (**D**) ^1^H-MRSI grid overlaid on T2-FLAIR image showing different colored voxels (yellowish and reddish: CRLB ≤ 40%; greenish and blueish: CRLB ≥ 40% for detecting 2HG). (**E**) Spectra from red voxel encompassing the tumor (white arrow) demonstrates elevated Cho/Cr (0.79; CRLB = 3%) and metabolic levels of 2HG (2HG/Cr =1.05; CRLB = 16%, black arrow). Histopathological and immunohistochemical analyses were consistent with grade-3 astrocytoma with positive *IDH*1 mutational status. Red lines: fitting spectra. Black lines: real spectra.

**Figure 6 cancers-15-04453-f006:**
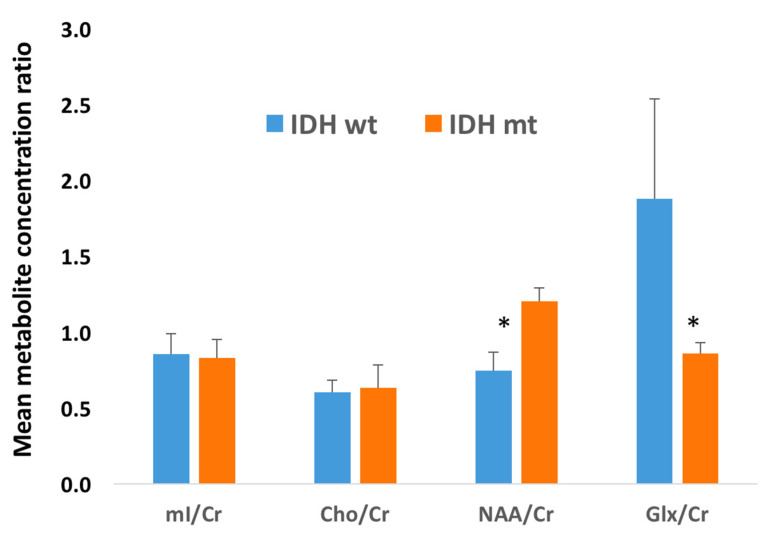
Bar graphs represent the metabolite concentration ratio as mean ± SEM (standard error of the mean). Non-parametric Mann–Whitney U tests were performed to ascertain the differences in metabolite ratios between two groups. The symbol (*) indicates significant difference (*p* < 0.05). The *IDH*-mutant gliomas had a significantly higher mean concentration of NAA/Cr ratio (1.20 ± 0.09 versus 0.75 ± 0.12; *p* = 0.016) compared with *IDH*-wild-type patients and significantly lower Glx/Cr ratio (0.86 ± 0.08 vs. 1.88 ± 0.66; *p* = 0.029) than *IDH-*wild-type gliomas.

**Table 1 cancers-15-04453-t001:** Patient demographics, histological/immuno-histochemical findings, and ^1^H-MRS results.

Patient	Gender	Age (Years)	^1^H-MRS Modality	^1^H-MRS 2HG Results	*IDH*1 Status	Histopathology Results	Status at Time of ^1^H-MRS
1	F	54	SVS	Negative	*IDH*1-R132H negative	Glioblastoma, WHO grade 4	New
2	M	63	SVS	Positive	*IDH*1-R132H mutant	Oligodendroglioma, WHO grade 2	New
3	F	32	SVS	Positive	*IDH*1-mutant	Mixed Oligoastrocytoma, WHO grade 3	Recurrent
4	F	34	SVS	Positive	*IDH*1-R132H mutant	Astrocytoma, WHO grade 3	New
5	M	38	SVS	Negative	*IDH*1-R132H negative	Glioblastoma (RTK1 subclass), WHO grade 4	New
6	F	53	SVS	Negative	*IDH*1-R132H negative	Glioblastoma, WHO grade 4	New
7	F	72	SVS	Negative	*IDH*1-R132H negative	Molecular Glioblastoma, WHO grade 4	New
8	M	24	SVS	Positive	*IDH*1-R132H mutant	Anaplastic Astrocytoma, WHO grade 3	New
9	F	36	SVS	Positive	*IDH*1-R132H mutant	Astrocytoma, WHO grade 3	New
10	F	48	SVS	Positive	*IDH*1-R132H mutant	Astrocytoma, WHO grade 3	New
11	F	46	SVS	Positive	*IDH*1-R132H mutant	Oligodendroglioma, WHO grade 2	New
12	M	64	SVS	Negative	*IDH*1-R132H negative	Glioblastoma, WHO grade 4	New
13	F	28	^1^H-MRSI	Positive	*IDH*1-R132H mutant	Astrocytoma, WHO grade 3	New
14	M	36	^1^H-MRSI	Positive	*IDH*1-R132H mutant	Anaplastic Astrocytoma, WHO grade 3	New
15	M	69	^1^H-MRSI	Negative	*IDH*1-R132H negative	Anaplastic Astrocytoma, WHO grade 3	New
16	F	40	^1^H-MRSI	Positive	*IDH*1-R132H mutant	Anaplastic Astrocytoma, WHO grade 3	New
17	F	36	^1^H-MRSI	Positive	*IDH*1-R132H mutant	Anaplastic Oligoastrocytoma, WHO Grade 3	New
18	M	39	^1^H-MRSI	Negative	*IDH*1-R132H negative	Glioblastoma, WHO grade 4	New
19	M	53	^1^H-MRSI	Negative	*IDH*1-R132H negative	Glioblastoma with sarcomatous features, WHO Grade 4	New
20	M	35	^1^H-MRSI	Positive	*IDH*1-mutant	Oligodendroglioma, WHO grade 2	New
21	M	30	^1^H-MRSI	Positive	*IDH*1-mutant	Anaplastic astrocytoma, WHO Grade 3	New
22	M	38	^1^H-MRSI	Positive	*IDH*1-R132H mutant	Diffuse Astrocytoma, WHO Grade 2	New
23	F	51	^1^H-MRSI	Positive	*IDH*1-mutant	Recurrent Astrocytoma, progression to WHO grade 4	Recurrent
24	F	80	^1^H-MRSI	Negative	*IDH*1-R132H negative	Infiltrating astrocytoma, WHO grade 2	New

^1^H-MRS = proton MR spectroscopy; ^1^H-MRSI = proton magnetic resonance spectroscopic imaging; 2HG = 2-hydroxyglutarate, *IDH* = Isocitrate dehydrogenase; New= newly diagnosed gliomas; SVS = single-voxel spectroscopy.

**Table 2 cancers-15-04453-t002:** Diagnostic performance of 2HG for prediction of *IDH*-mutant glioma.

		2HG Status	
		Negative	Positive
***IDH* Status**	**Wild-Type**	7	2
	**Mutant**	3	12
Accuracy (%)		79	
Sensitivity (%)		80	
Specificity (%)		77	
Positive Predictive Value (%)		86	
Negative Predictive Value (%)		70	

## Data Availability

The data presented in this study are available on request from the corresponding author.

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
