# Peer review of "Non-Invasive Assessment of Isocitrate Dehydrogenase-Mutant Gliomas Using Optimized Proton Magnetic Resonance Spectroscopy on a Routine Clinical 3-Tesla MRI"

_cancers, 2023, doi:10.3390/cancers15184453_

Round 1
Reviewer 1 Report
The authors have demonstrated in a small cohort of patients with suspected infiltrative gliomas (24 patients) that Proton Magnetic Resonance Spectroscopy may detect the presence of 2-hydroxyglutarate (2HG), indirect evidence of IDH mutation.
With an overall accuracy of 75% clearly this test cannot be used, at the present time, to inform significant clinical decision nor to make robust prognostic statements.
The authors describe in details the reason underlying these pitfalls, such as small tumor volume or non neomorphic activity of IDH mutation.
Perhaps conversion of 2HG positive tumor to non 2HG positive tumor may be an indication of further malignant development in a tumor followed up longitudinally with this technique. Of course this would also need to be supported by immuno histochemistry.
Author Response
POINT-BY-POINT RESPONSE TO REVIEWERS
Reviewer 1
The authors have demonstrated in a small cohort of patients with suspected infiltrative gliomas (24 patients) that Proton Magnetic Resonance Spectroscopy may detect the presence of 2- hydroxyglutarate (2HG), indirect evidence of IDH mutation. With an overall accuracy of 75% clearly this test cannot be used, at the present time, to inform significant clinical decision nor to make robust prognostic statements. The authors describe in details the reason underlying these pitfalls, such as small tumor volume or non neomorphic activity of IDH mutation. Perhaps conversion of 2HG positive tumor to non 2HG positive tumor may be an indication of further malignant development in a tumor followed up longitudinally with this technique. Of course this would also need to be supported by immuno histochemistry.
Response: We thank the reviewer for his comments and raising pertinent questions/concerns. We would like to clarify that a total of 32 patients suspected of infiltrative gliomas based on routine MRI findings were initially recruited in this study. However, eight cases were excluded as five patients had sub-optimal spectral quality due to inadequate water suppression for reliable detection and quantification of metabolites, and three other patients were excluded because they had final histological diagnoses of non-tumor etiology. As a result, 24 patients were included in the final data analysis. All these 24 patients had the availability of histopathological confirmation for infiltrative gliomas and immunohistochemistry analyses for IDH mutation. Our results indicate that 19/24 patients were correctly identified as IDH-mutant or IDH-wildtype gliomas by proton MR spectroscopy (TE=97ms) with an overall concordance rate of 79%. The sensitivity, specificity, positive predictive value, and negative predictive value were 80%, 77%, 86%, and 70%, respectively. We acknowledge that diagnostic performance of proton MR spectroscopy in our study is not very high, however, as you correctly mentioned, not all IDH-mutant gliomas, especially the non-canonical IDH mutant gliomas (about 20-25% of grade-2 and 5-12% of grade-3 gliomas), show the neomorphic activity of 2HG production. In addition, there are some constraints related to the tumor volume and voxel size, as described in the Discussion section. We have added a paragraph highlighting the potential advantages/disadvantages of single voxel and multivoxel spectroscopic methods under the ‘Discussion’ section (page 21) and the appropriate use of these techniques according to the specific tumor's features.
Regarding the second part of your comment, the evidence of change in IDH mutation over time is limited in the literature and seems to be rare and most likely related to the technical issues associated with immunohistochemistry analyses. However, the assessment of 2HG by proton MR spectroscopy can be used longitudinally to differentiate tumor recurrence from treatment-related changes, treatment monitoring and treatment assessment to standard and targeted therapies (IDH mutant-inhibitor drugs). In short, the growing body of evidence shows that 2HG-MRS with an optimized echo time may be useful in non-invasively predicting IDH mutational status, which has significant clinical implications for prognostication and clinical management, including surgical planning. We agree that our findings warrant further validation in large-scale and multicenter prospective studies, and we have addressed this issue in a separate limitation section (page 23).

Reviewer 2 Report
The authors present an interesting approach on the preoperative assessment of IDH mutation in glioma using MR spectroscopy. The manuscript is well structured and contains all necessary introducing and methodological information. However, there are some aspects that should be addressed and discussed.
- Why was in some cases only SVS, in some cases only HMRSI (which is actually CSI, chemical shift imaging), and why in some cases SVS and HMRSI used? How was it decided? As it is a prospective study acquiring SVS and HMRS might a good way also to compare the applicability of both modalities.
- Why were different voxel sizes chosen? Too small voxel sizes lead to noisy data, to large ones to partial volume effects. For comparability across subjects it would been beneficial to keep it constant.
- Eight outer volumes saturation slabs were used. Please specify (left/right, anterior/posterior, cranial/caudal?)
- Please add a Figure showing SVS and HMRSI planning and how the voxel of the SVS or grid of HMRSI was placed according to the lesion.
- Were tissue samples collected in a way, the spatial localization within the image data is known?
- How do SVS and HMRSI compare? Which fits better to the clinical needs? Can the authors provide some “guidelines” how to continue with this research?
Moderate editing needed
Author Response
POINT-BY-POINT RESPONSE TO REVIEWERS
We thank the reviewer for the thorough review of our manuscript. We acknowledge that all your feedback, both negative and positive; have been extremely helpful in improving the overall quality of the manuscript. All sentences or words added to the text are highlighted in BLUE. Words removed from the text are highlighted in YELLOW.
Reviewer 2
The authors present an interesting approach on the preoperative assessment of IDH mutation in glioma using MR spectroscopy. The manuscript is well structured and contains all necessary introducing and methodological information. However, there are some aspects that should be addressed and discussed.
Response: The authors appreciate the reviewer’s positive comments and favorable feedback. Please find below our responses to your individual concerns/suggestions.
1) Why was in some cases only SVS, in some cases only HMRSI (which is actually CSI, chemical shift imaging), and why in some cases SVS and HMRSI used? How was it decided? As it is a prospective study acquiring SVS and HMRS might a good way also to compare the applicability of both modalities.
Response: Thank you, reviewer, for your comment. The use of SVS was preferred for focal well-circumscribed more ‘superficial’ lesions and 1H-MRSI for ill-defined, irregular heterogeneous and larger lesions to avoid incomplete sampling of neoplasms. Because of time constrains, we used both techniques only in a few cases (n=3). Wherever it was feasible, we got similar results (2 cases with 2HG positive findings and one case with 2HG negative finding, in agreement with the immunohistochemical findings). This information has now been included in the ‘Results’ section (page 14). In addition, the accuracy of SVS (75%) and 1H-MRSI (83%) were fairly similar. We added a sentence in the ‘Discussion’ section (5th paragraph, page 21) describing the comparable potential of both techniques in predicting IDH-mutant gliomas from our patient cohort.
2) Why were different voxel sizes chosen? Too small voxel sizes lead to noisy data, to large ones to partial volume effects. For comparability across subjects it would been beneficial to keep it constant.
Response: Even though it would be beneficial to keep the voxel size constant across all subjects, MRS is highly dependent on tumor size, geometry and location, hence voxel size is usually variable in accordance to tumor’s features as well as to minimize acquiring spectroscopy data with poor signal to noise ratio and partial volume averaging effects (doi:10.1093/neuonc/noy113). Therefore, the SVS voxel size varied from 10 x 10 x 10 mm3 (volume: 1 cm3) – 20 x 20 x 20 mm3 (volume: 8 cm3) amongst our cohort. This information has been added to the Methods section (page 7). Our results are in line with de la Fuente MI, et al. (doi: 10.1093/neuonc/nov307), where it was also observed that diagnostic performance was higher for large tumors (>8 mL) than for small tumors (<3.4 mL), and with Izquierdo-Garcia JL, et al. (doi: 10.1371/journal.pone.0118781), where they reported higher detection sensitivity of 2HG from larger voxel size.
3) Eight outer volumes saturation slabs were used. Please specify (left/right, anterior/posterior, cranial/caudal?)
Response: The saturation slabs were applied along multiple directions, depending on tumor location, morphology, and orientation (see new Figure 2). We have included this information in the Methods section (page 8).
4) Please add a Figure showing SVS and HMRSI planning and how the voxel of the SVS or grid of HMRSI was placed according to the lesion.
Response: As per reviewer’s suggestion, we have added a modified figure (see new Figure 2 on page 8).
5) Were tissue samples collected in a way, the spatial localization within the image data is known?
Response: Although we did not perform a direct tissue sampling using imaging/spectroscopy-guided tumor resection or biopsy, the histopathological analyses revealed confirmed diagnosis of infiltrative gliomas from tumor specimens collected following complete or near total surgical resection of the hyperintense expansile mass as visible on T2-FLAIR images (this information included in the Data Processing sub-section (page 9). Since volume of interest was selected to include neoplasm visible as a hyperintense mass on T2-FLAIR images while acquiring MRS data, we believe that the tumor specimen used for histopathological and /or immunohistochemical analyses corresponds to the spatial localization within the MRS data.
6) How do SVS and HMRSI compare? Which fits better to the clinical needs? Can the authors provide some “guidelines” on how to continue with this research?
Response: We found a slight advantage of 1H-MRSI over SVS (accuracy 83% vs. 75%, respectively). However, each method has its own potential clinical benefits depending on the tumor characteristics. To emphasize this point, we have added a paragraph highlighting the potential advantages/disadvantages of single voxel and multivoxel spectroscopic methods under the ‘Discussion’ section (page 21).
“SVS is a broadly available method, generally providing a better signal-to-noise ratio especially from focal well-circumscribed lesions. Moreover, SVS is an easy to acquire sequence that allows good magnetic field shimming over the limited volume of interest and takes shorter acquisition time. On the other hand, 1H-MRSI is a valuable technique for obtaining metabolic information from ill-defined, irregular, heterogeneous, large(er) masses including the entire tumor volume to avoid incomplete sampling. Additionally, 1H-MRSI method allows acquisition of spectroscopy data with better spatial resolution, thus minimizing the partial volume averaging effects. This in turn allows metabolic mapping of tumor heterogeneity with lesser degree of admixing the signals from different tissue compartments. We believe that both methods are equally good when used appropriately according to the specific tumor's features. However, 1H-MRSI takes longer time to acquire the data and is harder to shim well over the entire region and prone to voxel bleeding effect resulting in signal contamination from lipids originating from scalp and bone marrow.”

Reviewer 3 Report
Dear Authors,
In general, the paper follows an adequate structure and correct scientific support and can be published considering some limitations.
Also, the study is interesting in the field.
However, there are a series of limitations that should be considered.
Abstract. Incorporate in the summary: a more precise sentence of the conclusions.
Introduction. This section presents the problem in a coherent and clear manner with the correct support of the scientific literature. One of the main objectives of research is to find out solutions to questions through the application of scientific methods.
2. Methods
Figure 2:
N = population
n = studies/sample/participants / cases
Please, insert n
Methods
Subjects
“The study was approved by the Institutional Review Board of the University of Pennsylvania”
Please, insert the Research Ethics Committee Reference Number.
Please, insert the Declaration of Helsinki in relation to Medical Research.
Please, insert the Oviedo Convention - This Convention is the only international legally binding instrument on the protection of human rights in the “biomedical” field.
Discussion
“Our results demonstrated that 1H-MRS can identify 209 IDH-mutant gliomas with high accuracy. (…) “Our results” also provided evidence that IDH (…)
Please, create content in a simple writing style.
Conclusion. Differentiate the discussion of the main conclusions of the study and modify the limitations of the study and locate them in said section at the end.
Finally, please, highlight the main contributions of the study.
References
They should be reviewed and updated according to the publication standards. There are many errors in the references.
For instance:
25. Franceschi E, De Biase D, Di Nunno V, et al. IDH1 Non-Canonical Mutations and Survival in Patients with Glioma. Diagnostics 345 (Basel) 2021;11.
48. Andronesi OC, Arrillaga-Romany IC, Ly KI, et al. Pharmacodynamics of mutant-IDH1 inhibitors in glioma patients probed by 391 in vivo 3D MRS imaging of 2-hydroxyglutarate. Nat Commun 2018;9:1474.
Thanks
Kind regards
-
Author Response
POINT-BY-POINT RESPONSE TO REVIEWERS
We thank the reviewer for the thorough review of our manuscript. We acknowledge that all your feedback, both negative and positive; have been extremely helpful in improving the overall quality of the manuscript. All sentences or words added to the text are highlighted in BLUE. Words removed from the text are highlighted in YELLOW.
Reviewer 3
In general, the paper follows an adequate structure and correct scientific support and can be published considering some limitations. Also, the study is interesting in the field. However, there are a series of limitations that should be considered.
Response: The authors appreciate the reviewer’s positive comments. Please find below our responses that address the other concerns:
1) Abstract. Incorporate in the summary: a more precise sentence of the conclusions.
Response: Thank you, reviewer, for your suggestion. In the revised draft of manuscript, we have included a new and more precise conclusory statement (page 3): "Noninvasive optimized 1H-MRS can accurately predict IDH mutational status and 2HG can serve as a valuable diagnostic and prognostic biomarker in patients with gliomas."
2) Introduction. This section presents the problem in a coherent and clear manner with the correct support of the scientific literature. One of the main objectives of research is to find out solutions to questions through the application of scientific methods.
Response: The authors appreciate the reviewer’s positive comments and favorable feedback regarding the ‘Introduction’.
3) Methods
Figure 2:
N = population
n = studies/sample/participants / cases
Please, insert n
Response: Thank you, reviewer, for noticing it. We replaced "N" with "n" throughout the flowchart (page 12).
4) Methods
Subjects
“The study was approved by the Institutional Review Board of the University of Pennsylvania”
Please, insert the Research Ethics Committee Reference Number.
Please, insert the Declaration of Helsinki in relation to Medical Research.
Please, insert the Oviedo Convention - This Convention is the only international legally binding instrument on the protection of human rights in the “biomedical” field.
Response: As per the reviewer’s suggestions, the following statements were included in the Methods section under the subheading, “Subjects” (page 6):
“Informed consent was obtained from each participant, and the research project was approved by the Institutional Review Board/Ethics Committee of the University of Pennsylvania (protocol # 829645). The study was designed and conducted according to the guidelines of the Declaration of Helsinki.
5) Discussion
“Our results demonstrated that 1H-MRS can identify IDH-mutant gliomas with high accuracy. (…) “Our results” also provided evidence that IDH (…)
Please, create content in a simple writing style.
Response: Thank you reviewer for your suggestion. We have rephrased the statement in a simple writing style (page 17): " Our results showed that 1H-MRS can identify IDH-mutant gliomas with high accuracy (79%), sensitivity (80%) and specificity (77%). We also found that IDH mutation affects other metabolic pathways leading to variations in metabolite pools of Glx and NAA.”
6) Conclusion. Differentiate the discussion of the main conclusions of the study and modify the limitations of the study and locate them in said section at the end.
Finally, please, highlight the main contributions of the study.
Response: We appreciate the reviewer's suggestion. Towards the end of manuscript, we have added a sub-section "Limitation" in which have described some potential shortcomings of our study. We have also included a separate paragraph before the "Limitation" sub-section highlighting the main contributions of the study (pages 21/22): "Our findings may aid in establishing 2HG as a surrogate marker of IDH mutational status in gliomas. Using an optimized TE of 97ms, both SVS and multi-voxel 1H-MRS methods are equally efficient in detecting complex resonances of 2HG. Other physiologically sensitive metabolites such as Glx and NAA may also serve as potential biomarkers for detecting IDH mutant gliomas. We believe that diagnostic performance of 1H-MRS in the identification of IDH mutational status in gliomas can be further improved by using an integrated approach of analyzing metabolite levels of 2HG, NAA, and Glx together in future studies”.
7) References
They should be reviewed and updated according to the publication standards. There are many errors in the references.
For instance:
- Franceschi E, De Biase D, Di Nunno V, et al. IDH1 Non-Canonical Mutations and Survival in Patients with Glioma. Diagnostics 345 (Basel) 2021;11.
- Andronesi OC, Arrillaga-Romany IC, Ly KI, et al. Pharmacodynamics of mutant-IDH1 inhibitors in glioma patients probed by 391 in vivo 3D MRS imaging of 2-hydroxyglutarate. Nat Commun 2018;9:1474.
Response: We thank the reviewer for noticing some formatting issues in the references. We have now revised all the references once again to make sure that they are cited according to the publication standards.
Round 2
Reviewer 2 Report
Thank you for the adaptions of the manuscript
minor spelling issues need to be adressed
Reviewer 3 Report
Dear Authors,
Thanks!
Kind Regards
-